# Multi-Environment and Multi-Year Bayesian Analysis Approach in *Coffee canephora*

**DOI:** 10.3390/plants11233274

**Published:** 2022-11-28

**Authors:** André Monzoli Covre, Flavia Alves da Silva, Gleison Oliosi, Caio Cezar Guedes Correa, Alexandre Pio Viana, Fabio Luiz Partelli

**Affiliations:** 1Centro Universitário Norte do Espírito Santo, Universidade Federal do Espírito Santo, Rodovia BR-101, Km 60, Litorâneo, São Mateus 29932-540, ES, Brazil; 2Laboratório de Melhoramento Genético Vegetal, Universidade Estadual do Norte Fluminense Darcy Ribeiro, Avenida Alberto Lamego, 2000, Campos dos Goytacazes 28013-602, RJ, Brazil; 3Departamento de Agronomia, Universidade Federal do Espírito Santo, Alto Universitário, S/N Guararema, Alegre 29500-000, ES, Brazil

**Keywords:** coffee production, Markov chain, informative priors, cultivar recommendation

## Abstract

This work aimed to use the Bayesian approach to discriminate 43 genotypes of *Coffea canephora* cv. Conilon, which were cultivated in two producing regions to identify the most stable and productive genotypes. The experiment was a randomized block design with three replications and seven plants per plot, carried out in the south of Bahia and the north of Espírito Santo, environments with different climatic conditions, and evaluated during four harvests. The proposed Bayesian methodology was implemented in R language, using the MCMCglmm package. This approach made it possible to find great genetic divergence between the materials, and detect significant effects for both genotype, environment, and year, but the hyper-parametrized models (block effect) presented problems of singularity and convergence. It was also possible to detect a few differences between crops within the same environment. With a model with lower residual, it was possible to recommend the most productive genotypes for both environments: LB1, AD1, Peneirão, Z21, and P2.

## 1. Introduction

Worldwide, more than 80 tropical countries employ 25 million farmers that produce around 175 million bags of coffee every year. Commonly, *Coffea arabica* represents 60% and *C. canephora* 40% of the annual revenue of 172 million USD [1]. Brazil is an important player, due it is the main producer and exporter in the world, with an estimated increase in global demand in the 2021/22 cycle due to the advance of control of COVID-19 in important consumer centers, such as Europe and the United States [2]. In this scenario of strong *C. arabica* valuation due to the significant loss of production in Brazil (−25%), the industry tends to increase demand for *C. canephora* coffee to reduce the cost of producing blends. The Espírito Santo and Bahia states in Brazil are the largest producer with almost 17.5 thousand bags of already processed coffee, with an increase of 12% of *C. canephora* in the last harvest [2]. In this context, this is an important crop for the surrounding regions mainly due to the local climate that exerts a strong influence on production.

The climate in the region is predominantly within the ideal limits for the culture, mainly *C.canephora* which is temperature tolerant up to 37 °C by photoprotective and antioxidant mechanisms [3]. Outside the limits from 17 to 31 °C its development and production are reduced or impaired [4,5]. Although the limits are wide, it has been observed that coffee is highly responsive to small variations in temperature [6]. Breeding programs have developed new coffee genotypes, providing not only desirable agronomic characteristics, but also greater adaptability to the environment [7].

Even though breeding programs develop better genotypes available to environments, they cannot be tested in many locations. This impossibility is mainly due to the inherent characteristics of perennial plants, which have a long reproductive cycle, a high annual variation, differences in precocity between genotypes, and productive longevity [8]. So, biometric studies that fill these gaps are important to identify productive cultivars adapted to the environment, which is the main component of the competitiveness of coffee plantations. Difficulties inherent to perennial crops such as coffee, in addition to the lack of information aimed at important regions, also pose problems for statistical models [9].

The challenge for statistical modeling is focused on the quantity and quality of data obtained by this type of crop, which certainly reduces the predictive power of the models. Both from the point of view of genetic breeding and commercial orchards, this consequence comes from the use of the same genotype for several years, a low survival rate during the life of the orchards, and the need for repeated evaluations in each individual over time [9]. Therefore, using a more accurate modeling approach can undoubtedly save resources and improve the chance of success of orchards with perennial plants.

An approach that has been used more frequently in these types of problems is the Bayesian approach. This approach is widely used with genomic data in culture [10,11,12], but for phenotypic data, it has already been successfully applied to other perennial plants [13,14,15]. The Bayesian approach enables advantages compared to the frequentist (REML/BLUP). The main one is the possibility of using previous information (called informative priors) incorporated in the parameter’s distribution of the model [16]. In the frequentist’s approach, previous data are generally used in a joint analysis, which often generates hyper-parameterized models, with unplanned/incompatible experimental design or missing observations. Usually, it’s become a source of variation in the model and does not add much information, beyond the possibility of identifying if the previous data had different averages/variances from the current experiment. Another advantage is that the credibility intervals tend to be smaller than the confidence intervals with a proper prior [17]. Given the above, this work aims to use the Bayesian approach to discriminate 43 genotypes of *C. canephora* genotypes cv. Conilon is the most widely cultivated cultivar in Brazil for the Robusta type of coffee in the main coffee-producing regions.

## 2. Results

Both genotypes, locations, and year effects incorporated into the model were significant (*p* < 0.05) by chi-square (Table 1). The models that include the block effect, in addition to increasing the number of parameters in the model, presented problems of singularity and convergence. It is also noted that the deviance of model 3 + 5 (which corresponds to the 3rd model plus the block effect) is close to model 3, thus, the block effect was disregarded in the models. The model chosen was the 3 + 8 model, in which the year models both the intercept and slope of the location, at the cost of one more parameter in the model (six to seven), but with less deviance.

After choosing the model, it was adjusted following a Bayesian approach. The chain generated by the approach passed on the quality statistics of the Markov Chain convergence and sample size tests (Table A1). In addition to the tests, all chains were visually inspected (Figure 1 and Figure A1, Figure A2, Figure A3, Figure A4 and Figure A5), looking for initial instability in the chain that could be due to a small burn-in or problems along the chain that could come from either the model or inadequate parameters in the prior. The distribution of residues was also checked to identify any possible bias in the model or non-incorporated effect (Figure A6). No issues were found in either case. Observing the chains from parameters year inside the environment (Figure 1) of the genotypes in each year and location can be a good indication of how the genotypes are responding in these environments.

A biannually coffee yield was observed in Bahia, but it was not observed in Espírito Santo (Figure 1). Despite this biannually, the average yield in Bahia (100.35 bags of green coffee·ha^−1^) was not significantly different from Espírito Santo (84.05 bags of green coffee·ha^−1^), as well as both farmers who produce coffee in the two locations have the potential to produce similar amounts. In the years 2017 and 2018 in Bahia, the highest average production of coffee genotypes was observed, with 133.58 (lower CI = 126.60 and upper CI = 140.40) and 126.02 (lower CI = 118.96 and upper CI = 132.74) bags of green coffee·ha^−1^. These two years were significantly higher than the others evaluated in Bahia. The averages and credibility intervals observed in Espírito Santo did not show significant differences between the years in that location. Subsequently, some variance components were estimated from the genotype chains, mainly obtaining heritability and genotype ranking by BLUP (Table 2).

Finally, a ranking of genotype production was obtained, allowing farms in these locations to choose the most productive genotypes among those evaluated for use in their orchards (Figure 2). The heritability of the yield trait was about 0.28 with credibility intervals between 0.18 and 0.37.

## 3. Discussion

Accurate estimates and production pattern for a crop is important for the regional development of the crop, mainly because it allows accurate planning and profit forecasting on the cultivation activity. This importance is easily seen when a farmer must choose between two slightly different locations to plant their orchard. If initial studies do not consider that there may be a different variance in yield between the two environments and obtain a single average estimate for both, yield may be reduced by some non-incorporated effect (such as unfavorable weather). This loss can be further attenuated if considered an orchard destined for export, which generally yields 40% more than conventional ones.

The coffee crop is planted in a relatively wide spacing, where to obtain an orchard with a few hundred plants, the producer easily needs a large area. Thinking about the size needed to experiment, it was planned to be installed in a block design, as we believe that the soil might not be very homogeneous. After collecting all the data and thinking about the dependent random variable and all the explanatory variables, we tried to keep the model as inflated as possible. Later, use some form of model selection to choose a model that balances predictive accuracy and overfitting/Type I error [18,19], removing only the terms necessary to allow for a non-singular adjustment [20]. So, looking at our model simplification results, mainly looking at how much each factor contributes to reducing the model’s uncertainty, we believe that the block effect can be ignored. This factor contributed little to a better fit of the model, in addition to causing convergence problems.

After simplifying the model, we chose to use a Bayesian approach. The Bayesian approach has some advantages over frequentist analysis. The main one considered in our choice is the possibility of using informative priors on model parameters [16]. After several years of following farms and collecting data in this important coffee-producing region, in some cases with the same genotypes, we could include this prior information as a joint experiment for example. This would generate unbalanced data, sources of variation without enough repetitions, hyper parametrization, complexity, and perhaps increase model uncertainty. Another advantage is that credibility intervals are often reported as smaller than confidence intervals if an appropriate prior has been used, as observed in guava, another perennial fruit tree [14]. Due to the likelihood function, if a non-informative prior is used, a Bayesian performance at least similar to the frequentist is expected [21,22,23].

The observation of the aspect of the chain is also a quality control criterion frequently used in the adjustment of the Bayesian model to the data [24]. If any problems such as chain instability, lack of convergence, or instability were observed, we could revise our prior, chain size, or even the sources of variation in the model. If there is interest from other breeders or biometrics, our chains, and their quality tests, as well as algorithms, can be obtained from Appendix A.

Disregarding the blocking effect, all others presented at least one significant contrast (*p* < 0.05), moreover, looking at the residuals of the model, we did not observe any bias. This implies that the effects we considered in the model, in addition to being significant, were reasonably sufficient to control the errors. This information is particularly important to discriminate which genotypes are more promising, and the farmer in the region of the greatest importance for the crop in Brazil can better plan the harvest volume, even in regions relatively close to those evaluated in the experiment. All the genotypes studied here had already been studied for other traits related to grain quality [25], and a significant difference was observed between the genotypes, like as differences between the growth of the root system, which is directly correlated with yield, nutrient concentrations in leaves, our main source of variation of interest for recommendation to farmers.

This fluctuation in production observed in Bahia has already been observed in another trait, this is directly related to the best rainfall conditions concerning the state of Espírito Santo, the leaf concentrations of macro- and micronutrients are at better levels [26]. The rainfall and temperature as well as the reproductive periods influenced this trait, and as here the crop has closed the cycle for all genotypes, we believe that climatic data are strong candidates to enter as covariates in future models in cultivar recommendation systems. Here, our year/location as random provides a hint that something could vary between the interaction of these variables, but we don’t know what it is, although a good guess would be the weather.

Although we did not detect a significant difference between the locations, in the meteorological data we observe that Bahia, for different periods, had an average temperature higher than Espírito Santo. This may have contributed to a greater sum of degree days in the culture at that location. On the other hand, a lack of consistency in Espírito Santo may have generated greater variability in production there, which led to greater credibility intervals than those observed in Bahia.

So, was proceeded with the rank of the individuals by BLUP obtained using the Bayesian approach. The Bayesian MCMC methods consider uncertainties in the parameters throughout the inference process. On the other hand, the frequentist models are predicted by punctual estimates of variance components and considered as true values, ignoring uncertainty in the variance parameters [27].

It was possible to observe great genotypic variability in the group of studied genotypes. The most relevant information here is the group of the first five genotypes, which stood out from the others and can be used by farmers who want high yields. The credibility intervals (HPD interval) found can be considered relatively high, but this great variability may come from the interaction between the environmental genotype, mainly in the Espírito Santo locations. This does not rule out the other genotypes as potential candidates to compose an orchard with characteristics that the most productive may not have, such as good stability between environments, and great responsibility to favorable environments. However, another study with the material is needed to discriminate these genotypes for these characteristics. The heritability found was also a value within the expected, which is not very high but is consistent with the polygenic nature of the studied trait. The five genotypes shown in Figure 2 are good recommendations for people interested in growing crops in the region, or any other environment with similar conditions, especially regarding altitude. The selection of the best families was performed to be recombined and to generate new populations.

## 4. Materials and Methods

### 4.1. Plant Material and Field Experiments

The genetic material consisted of 43 coffee (*C. canephora*) genotypes, both adapted to tropical environments, and widely cultivated by coffee farmers in the State of Espírito Santo—Brazil. The genotypes were evaluated for high-yield potential and agronomic traits of interest. The experiment was planted at a spacing of 3.5 m × 1.0 m, in a randomized block design with three replications, and seven plants as an experimental unit. All cultural treatments were made according to the crop requirement, and pruning kept around 12,000 orthotropic branches.

The trials were carried out in two environments in Brazil, in the south of Bahia (16°36′52″ S, 39°30′33″ W, and 140 m asl) and the north of Espírito Santo (18°39′43” S, 40°25′52″ W, and 200 m asl). According to Köppen’s classification, the regional climate is Aw tropical, with hot humid summers and dry winters, mean minimum temperatures of over 15 °C (July to August) and mean maxima of over 35 °C in January to February (Figure 3).

The edaphoclimatic conditions in these two locations characterize slightly different conditions, and therefore two distinct environments for the plants. At each location, four harvests were carried out, between the harvested period 2016 to 2019 at the experimental plot level, according to the maturation cycle of each genotype. The individual yield (bags of green coffee per hectare) was used as a random variable in the statistical analyses.

### 4.2. Model Choice

A statistical model was implemented that contemplates the initially planned experimental design, with all the identified effects. From this model called the complete model, a simplification of the model was performed, removing non-significative effects or their combinations. After that, a Bayesian approach was used. This simplification helps to reduce the number of model parameters and complexity, in addition to helping the processing time in Bayesian chains, reducing their numbers if there are fewer effects in the model. Complete model:(1)$$y_{\left(ijk\right)}=\beta_{0}+\beta_{1}g_{\left(i\right)}+\beta_{2}p_{\left(j\right)}+\beta_{3}{\left(p/t\right)}_{\left(jk\right)}+\beta_{4}{\left(p/b\right)}_{\left(jl\right)}+\epsilon_{\left(ijk\right)}\left(I\right)$$
where: $y_{\left(ijk\right)}$ is the vector of the phenotypic values of the yield in the $i^{th}$ genotype of $j^{th}$ location of $k^{th}$ year; $\beta_{0}$ is an associated model parameter (intercept); $g_{\left[i=1,\ldots ,43\right]}$ is the parametric vector of random effects of $i^{th}$ genotype associated with the vector y by the incidence matrix known $\beta_{1}$, assuming that $g∼N\left(0,I⊗\Sigma_{g}\right)$ $p_{\left[j=1,\ldots ,2\right]}$ is the parametric vector of fixed (systematic) effects of $j^{th}$ location associated with the vector *y* by the incidence matrix known $\beta_{2}$ assuming that $p∼N\left(\mu_{p},I⊗\Sigma_{p}\right)$;$t_{\left[k=1,\ldots ,4\right]}$ is the parametric vector of random effects of $k^{th}$ year inside $j^{th}$ location influencing both intercept and slope, associated with the vector y by the incidence matrix known $\beta_{3}$, assuming that $t∼N\left(0,I⊗\Sigma_{t}\right)$; $b_{\left[l=1,\ldots ,3\right]}$ is the parametric vector of random effects of $l^{th}$ block inside $j^{th}$ location influencing both intercept and slope, associated with the vector y by the incidence matrix known $\beta_{4}$, assuming that $t∼N\left(0,I⊗\Sigma_{l}\right)$; and ϵ is the vector of random residual effects, assuming that $\epsilon ∼N\left(0,I⊗\Sigma_{\epsilon }\right)$.

From this complete model(I), a simplification of the model was performed using *package::functionlme4::lmer* [18] in R language [28]. Were observed its goodness of fit through deviance and the number of parameters that each effect includes in the model. After simplification, a model for fit (II) using the Bayesian approach was chosen, using the same definitions for the effects as the full model described above:(2)$$y_{\left(ijk\right)}=\beta_{0}+\beta_{1}g_{\left(i\right)}+\beta_{2}p_{\left(j\right)}+\beta_{3}{\left(p/t\right)}_{\left(jk\right)}+\epsilon_{\left(ijk\right)}(\mathrm{II})$$

### 4.3. Bayesian Approach

The Bayesian approach was used with the same model (II), applying the Monte Carlo method based on Markov Chains (MCMC), employing the package *MCMCglmm::MCMCglmmas* [29], in R language. A total of 500,000 iterations were estimated, discarding the first one at 50,000 (*burn-in*) and performing a 1:5 (*thin*) sampling. Quality statistics of the Markov Chain convergence were tested by the Geweke criterion [30] using the *coda::geweke.diag* package [31] in R language.

The formula represented in model (II) can be written more compactly, representing the system of equations with random effects models, expressed as:(3)$$E\left[y\right]=X\beta +Z\mu +\epsilon (\mathrm{III})$$
where: X is the design matrix for fixed effects and contains the predictor information and $\beta =\left[\beta_{0},\beta_{2}\right]\prime$ is the vector of parameters; Z is the design matrix for random effects and $\mu =\left[\mu_{1},\mu_{3}\right]$ is the vector of parameters; and ϵ is the error associated with the model, assuming $y∼N\left(X\beta ,I⊗\Sigma_{\epsilon }\right)$.

In a Bayesian approach, both fixed and random effects are random, also called systematic effects, treated in the same way [29], and combine the design matrices $\left(W=\left[X,Z\right]\right)$ and combine the vectors of parameters $\left(\theta =\left[\beta \prime ,\mu \prime \right]\prime \right)$:(4)$$E\left[y\right]=W\theta +\epsilon (\mathrm{IV})$$

In this formula, the design matrix is exactly equivalent to the original design matrix (frequentist) except we have one additional variable. In the frequentist model, this variable is absorbed into the global intercept because it could not be uniquely estimated from the data. Now, in the Bayesian approach, this additional parameter can be estimated by inclusion from a prior. As is usual in a Bayesian analysis, if there is no information in the data it has to come from the prior.

Priors can be defined for the residuals, the fixed effects, and the random effects. *MCMCglmm::MCMCglmm* uses an inverse-Wishart distribution prior for the (co)variances matrices and a normal prior for the fixed effects [29]. For model (II), the prior information was based on the posterior distributions of the parameters from a meta-analysis of coffee experiments [32,33,34,35]. The prior informative probability distribution for the systematic parameters of interest was provided by: $\beta_{i}\sim N\left(\mu_{0},V_{b}\right)$ in which $\mu_{0\;}$ is a vector of averages and $V_{b}$ is a diagonal matrix of the a priori variance of β. Also was used an inverse-Wishart distribution for both random $G_{0}$ and residuals $R_{0}$ as a priori for the covariance matrices: $G_{0}\sim W_{1}^{-1}\left(\Sigma_{g},n\right)$ and $R_{0}\sim W_{1}^{-1}\left(\Sigma_{\epsilon },n\right)$ in which $\Sigma_{g}$ and $\Sigma_{\epsilon }$ are scale matrices.

From chains, the a posteriori distribution was used to take averages, credibility intervals, breeding values, variance components, and broad-sense heritability, using the expression: $h^{2}=\frac{\sigma_{g}^{2}}{\left(\sigma_{g}^{2}+\sigma_{p/t}^{2}+\sigma_{\epsilon }^{2}\right)}$, where: $\sigma_{g}^{2}$, $\sigma_{p/t}^{2}$ and $\sigma_{\epsilon }^{2}$ are the genotypic, year inside the environment, and residual variances, respectively.

## Figures and Tables

**Figure 1 plants-11-03274-f001:**
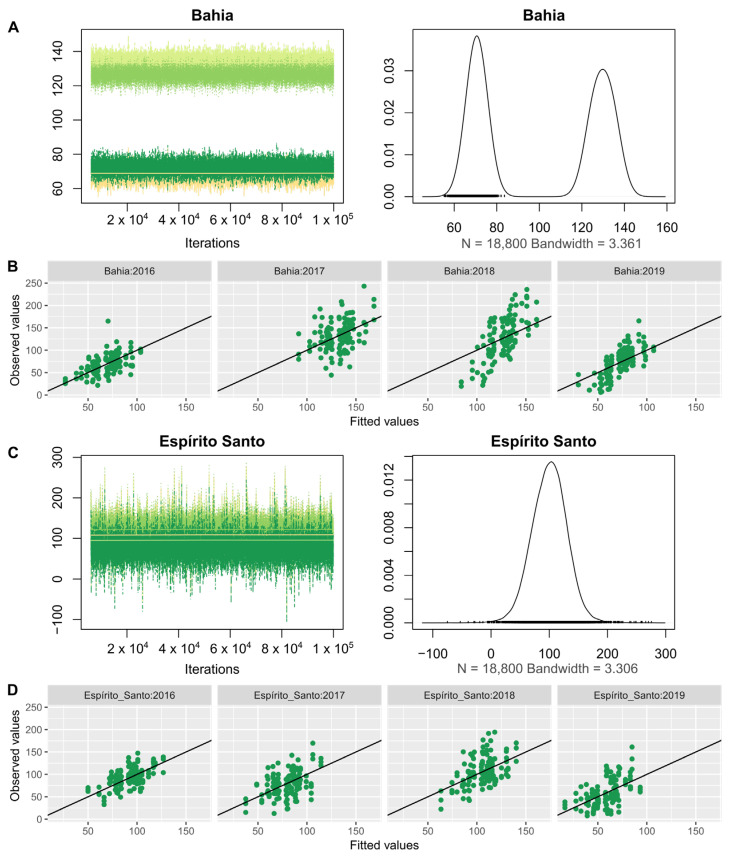
Time series of the chains of the four crops within the Bahia location, and probability density function curve of the chains (**A**). Observed versus predicted values for the four crops in the Bahia location (**B**). Time series of chains of the four crops within the Espírito Santo location, and probability density function curve of the chains (**C**). Observed versus predicted values for the four cultures at the Espírito Santo location (**D**).

**Figure 2 plants-11-03274-f002:**
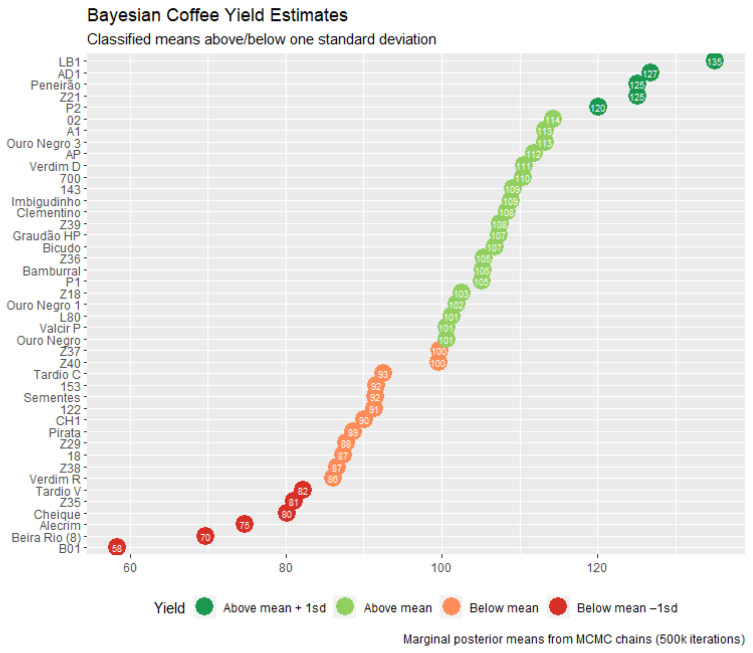
Yield average estimates for four harvests for 43 coffee (*Coffea canephora*) genotypes grown in Bahia and Espírito Santo. Means were grouped into values above one standard deviation from the mean, above the mean, below the mean, and below one standard deviation from the mean.

**Figure 3 plants-11-03274-f003:**
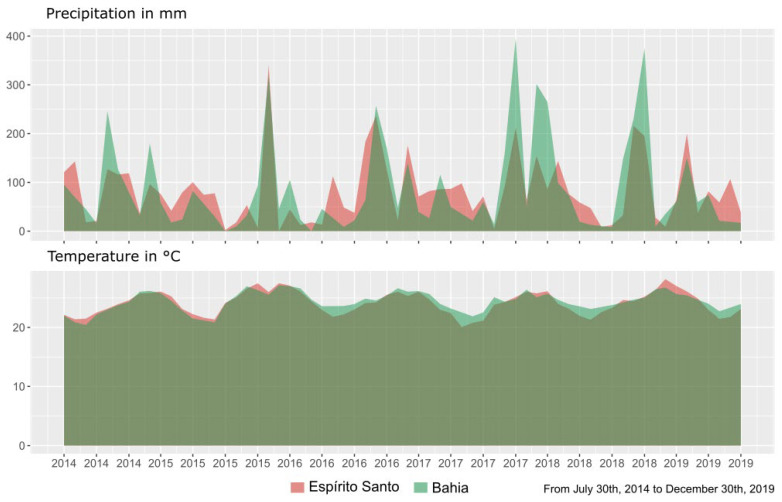
Information on monthly averages of total precipitation and the average temperature of climate data from two environments, Bahia and Espírito Santo, in the period between the implementation of the experiment and the last evaluated coffee crop (*C. canephora*).

**Table 1 plants-11-03274-t001:** Summary of ANOVA to simplify models with good-fit statistics on 43 coffee genotypes (*Coffea canephora*) for yield evaluated in two locations and four years.

	Model	Npar	DF	Chisq	Pr (>Chisq)	Deviance
1	Null	2	-	-	-	10,579.3
2	(1|genotype)	3	1	91.2	1.2 × 10^−21^	10,488.1
3	(1|genotype) + location	4	1	48.8	2.7 × 10^−12^	10,439.2
3 + 4	(1|year)	5	1	345.7	3.6 × 10^−77^	10,093.5
3 + 5	(1|block) *	5	0	0.0	-	10,438.6
3 + 6	(1|location/year)	6	1	557.7	2.5 × 10^−123^	9880.8
3 + 7	(1|location/block) *	6	0	0.0	-	10,438.6
3 + 8	(location|year)	7	1	559.6	1.0 × 10^−123^	9879.0
3 + 9	(location|block) **	7	0	0.0	-	10,437.1
3 + 10	(location|block) + (location|year) **	10	3	566.1	2.2 × 10^−122^	9871.0

Note: From the third model onwards, all consider the effect of genotype and location. * Model failed to converge. ** boundary (singular) fit. Npar = number of parameters. DF = degrees of freedom. Chisq = chi-square test.

**Table 2 plants-11-03274-t002:** The posterior mean of BLUP estimates for 43 coffee genotypes (*Coffea canephora*) and their higher posterior density (HPD) interval, estimates were obtained by a Bayesian model.

Genotype	BLUP	HPD Interval		Genotype	BLUP	HPD Interval	
		Lower	Upper			Lower	Upper
LB1 *^,#^	34.76	23.41	46.64	L80 ^#^	0.94	−10.45	12.41
AD1 *^,#^	26.40	14.52	37.79	Valcir P	0.31	−11.73	11.50
Peneirão *^,#^	24.81	13.27	36.13	Ouro Negro	0.27	−11.11	11.65
Z21	24.74	12.99	36.03	Z37	−0.63	−11.87	11.17
P2 *	19.72	7.93	31.13	Z40	−0.76	−12.16	10.63
02	13.90	2.17	25.28	Tardio C	−7.84	−19.00	3.91
A1 ^#^	12.93	1.55	24.62	153	−8.71	−19.98	3.05
Ouro Negro 3	12.92	1.59	24.73	Sementes	−8.83	−20.56	2.33
AP *	11.45	−0.17	22.85	122	−9.02	−20.37	2.65
Verdim D	10.23	−1.07	21.68	CH1	−10.25	−21.87	1.22
700	10.12	−1.30	21.48	Pirata	−11.70	−23.25	−0.05
143	8.75	−2.64	20.24	Z29	−12.58	−24.19	−1.29
Imbigudinho *	8.56	−2.96	20.06	18	−13.06	−24.36	−1.50
Clementino	7.96	−3.43	19.79	Z38	−13.73	−25.31	−2.40
Z39	7.16	−4.37	18.63	Verdim R	−14.28	−26.31	−2.92
Graudão HP	7.03	−4.67	18.30	Tardio V	−18.12	−29.96	−6.54
Bicudo ^#^	6.52	−5.08	17.78	Z35	−19.31	−30.96	−8.02
Z36	5.06	−6.04	16.77	Cheique	−20.16	−31.58	−8.34
Bamburral	4.97	−6.83	16.35	Alecrim	−25.66	−36.92	−13.85
P1	4.78	−6.58	16.21	Beira Rio (8)	−30.68	−42.06	−18.86
Z18	2.23	−8.71	14.05	B01	−41.98	−53.54	−30.46
Ouro Negro 1	1.54	−9.60	12.99	-	-	-	-

* Derived from the cultivar Monte Pascoal (Partelli et al., 2021), ^#^ derived from the cultivar Plena which is in the registration process.

## Data Availability

The complete phenotypic information, breeding values, and generated chains used in this study were submitted to the Open Science Framework and received the public identifier DOI: https://doi.org/10.17605/OSF.IO/D8T2R, accessed on 1 November 2022.

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
