# Peer review of "Multi-Environment and Multi-Year Bayesian Analysis Approach in Coffee canephora"

_plants, 2022, doi:10.3390/plants11233274_

Round 1
Reviewer 1 Report
The topic of the manuscript " Multi-environment and multi-year Bayesian analysis approach in Coffee Canephora " is timely and fits within the scope of the Journal. However, I have highlighted weakness in the manuscript that need improvement. This is table 1 (section 2). I provide detailed comments below.

Author Response
Thanks to the reviewer for their comments.
Reviewer's first comment: Table 1 is too large. AIC and BIC measures show the same information. LogLik values are not used in
the text. I suggest removing three columns: AIC, BIC and logLik.
Answer: We agree with reviewer that the information is redundant. We thought that they could be useful for some other researcher looking for some parameter to compare their results with our work. However, these parameters are relatively simple to obtain, and can be calculated using the information already available in the table. We chose to remove the columns from the table (line 88) as suggested, leaving only DEVIANCE.
Second point of the reviewer: Why the dynamic concept of stability has not been studied, i.e., those genotypes that did not show genotype-environment interaction were not identified? Genotype-year or genotype-place could be a genotype-environment interaction.
Answer: This concept was not really addressed in the work because we would like to focus on how other breeders could apply the technique shown, instead of showing results on the material studied, since in the example there are only two places and a few years. However, someone who is already able to use and interpret this concept through the frequentist approach will easily be able to arrive at the results through the approach presented here, since the concept is based on the change in magnitude between coefficients of the model between favorable and unfavorable environments. With the Bayesian approach, these same coefficients are obtained at the end, leaving it up to the biometrician to interpret them.
Reviewer 2 Report
It was a pleasure to read this excellent manuscript. I especially appreciated the choice of an informative prior, and the balance between predictive accuracy and overfitting (using AIC and BIC). I also liked the clear recommendation of five winners in the last sentence of the abstract. My suggestions are very minor.
Because the same genotype recommendations are given for both locations, the winners seem suitable for a large growing region. Do the authors think these winners are best throughout the coffee region of Brazil? Can they make plausible recommendations for coffee producers in other countries? If so, that would add interest to the results.
This manuscript discusses the literature on good bean quality traits. If this applies to the five winners in particular, this should be mentioned. High yield is desired, but only if there is no great sacrifice of other desirable traits.
Again, this is an excellent manuscript.
Author Response
We are very grateful for the reviewer's comments.
Reviewer's first point: As the same genotype recommendations are given for both locations, the winners seem suitable for a high-growth region. Do the authors think these winners are the best in the entire coffee region of Brazil? Can they make plausible recommendations for coffee producers in other countries? If so, that would add interest to the results.
Repply: In Brazil, coffee genotypes are intrinsically linked to the altitude where they are grown, so our winners recommendation is applicable to other regions that have similar altitudes (+-350 m), not only in Brazil, but in other countries. . As these genotypes are the most stable, production predictability is also a very important point for potential stakeholders. We added this information to the results to make it clearer for readers (line 216-219).
Second point: This manuscript discusses the literature on good bean quality traits. If this applies to the five winners in particular, this should be mentioned. High yield is desired, but only if there is no great sacrifice of other desirable traits.
Repply: We fully agree with the reviewer that production should not be considered at the expense of other quality characteristics of coffee beans. We really discussed this point, and we had previously mentioned "Some of the genotypes studied here had already been studied for other traits related to grain quality [25]" and now we update to "All the genotypes studied here had already been studied for other traits related to grain quality [25]". This is another study by our research group, which involved other academics, and after crossing the data we found that it does not apply only to the five winning genotypes. These works are complementary, and both enrich the results. Thanks to the reviewer for noting this.